# Perspectives, benefits and challenges of a live OSCE during the COVID-19 pandemic in a cross-sectional study

Teresa Loda [1], Rebecca Sarah Erschens [2], Andrew B Nevins,[3] Stephan Zipfel,[2,4] Anne Herrmann-Werner [1,2]

[1]Tübingen Institute for Medical Education, University of Tuebingen, Tuebingen, Germany
[2]Department of Psychosomatic Medicine and Psychotherapy, University Hospital Tubingen Department of Internal Medicine, Tubingen, Germany
[3]Department of Medicine, Stanford University School of Medicine, Stanford, California, USA
[4]Faculty of Medicine, Eberhard-Karls University of Tuebingen, Tuebingen, Germany

**Correspondence to**
Teresa Loda;
teresa.loda@med.uni-tuebingen.de

## ABSTRACT

**Objectives** Restrictions due to the COVID-19 pandemic mandated fundamental changes to student evaluations, including the administration of the observed structured clinical examination (OSCE). This study aims to conduct an in-person OSCE to verify students' practical skills under necessary infection control practices and the impact of face masks on student–patient interactions.

**Design** Cross-sectional design.

**Setting** The OSCE at Medical School of Tuebingen takes place in October 2020.

**Participants** A total of 149 students (third year of study) completed the survey (RR=80.1%). It was their first OSCE.

**Primary and secondary outcome measures** Primary outcome measure was how this type of OSCE was evaluated by participating students in regard to preparation, content and difficulty as well as in real life. Secondary outcome measures were how the implemented hygiene actions influenced the OSCE, including the interaction and communication between students and standardised patients (SPs). Items were rated on a 6-point Likert scale (1=completely to 6=not at all). Means, SDs, frequencies and percentages were calculated.

**Results** 149 students, 32 SPs and 59 examiners participated. The students rated the OSCE with 2.37 (±0.52) for preparation and 2.07 (±0.32) for content. They perceived the interaction to be significantly disrupted by the use of face masks (3.03±1.54) (p<0.001) compared with the SPs (3.84±1.44) and the examiners (4.14±1.55). In general, the three groups considered the use of face masking the OSCE to be helpful (1.60±1.15).

**Conclusions** An in-person OSCE, even in the midst of a global pandemic, is feasible and acceptable to both students and faculty. When compared the students' results to previous students' results who completed the OSCE before the pandemic, the results indicated that students felt less prepared than under non-pandemic circumstances; however, their performances on this OSCE were not lower.

## INTRODUCTION

The COVID-19 pandemic resulted in significant adaptations for teaching in medical schools, with a forced emphasis on digitalisation; for example, e-learning platforms, virtual training and videoconferencing have become established strategies for teaching

## STRENGTHS AND LIMITATIONS OF THIS STUDY

⇒ The study presents the evaluation of an in-person observed structured clinical examination (OSCE) in the height of this pandemic.
⇒ Best to our knowledge, this is the first investigation of the influence of face masks regarding communication and interaction in an OSCE.
⇒ The perspective of all people involved in the OSCE was assessed.
⇒ One limitation is that the students might feel more stressed during the OSCE as it is an examination.
⇒ The study was conducted at only one medical faculty.

students and others studying in the medical field.[1–4] Initially, traditional face-to-face teaching formats such as physical examinations had to be paused completely due to stringent governmental orders and the need to develop appropriate infection control practices.[5 6] This change in format had impacts on how to teach and what to teach, as some clinical skills, in particular communication skills training, are difficult to adapt from in-person to remote sessions. However, telemedicine has become a valuable option for teaching such clinical and communication skills.[7]

Despite new opportunities, this alternative method of teaching remains challenging, particularly the question of how to perform corresponding valid examinations under restrictions required to prevent the spread of COVID-19. Various approaches had to be tried to accommodate different phases of the pandemic itself.[1 8] Catalogues of learning objectives also had to be adapted accordingly, resulting in direct impacts on the content of various examinations. With great effort, examinations themselves had to be altered, either reformulated as online versions with all the related data protection implications or held in classrooms under strict hygienic measures. Clinical and communication skills examinations, in particular, were a highly significant

challenge and were sometimes omitted entirely during the first COVID-19 semester in summer 2020 as the risk of infection was too high. This especially affected the creation and administration of observed structured clinical examinations (OSCEs) with their interdisciplinary setting evaluated by teachers of various disciplines.[9] The OSCE represents a standard assessment tool for medical training and is considered the most reliable and valid clinical examination system in undergraduate training[10]; therefore, a way to implement the OSCE during the pandemic COVID-19 had to be identified. Several universities solved this problem by conducting virtual or online OSCEs using Zoom videoconferencing.[11–15] However, the degree of realism in the virtual OSCE is limited, and the OSCE's organisational team needed to be prepared to manage and resolve technical difficulties.[11]

When the curve of the COVID-19 pandemic flattened, the Medical Faculty of Tuebingen decided that courses including clinical examinations should go back to being held in person, maintaining appropriate and strict infection control practices since such clinical encounters represent an essential component of medical training. This particularly affected the OSCE as the faculty's central interdisciplinary clinical examination. In this context, appropriate hygiene measures were developed by the OSCE coordination team based on the university and government specifications in order to minimise infection risk for all persons involved in the OSCE including students, examiners and standardised patients (SPs). Here, we describe our successes in this effort and several of the challenges we faced.

### Aim

This study aims to present one method for facilitating an OSCE during a real-world pandemic situation. Thus, as primary aim, this study investigated how this type of OSCE was evaluated by participating students in regard to preparation, content and difficulty as well as in real life. Furthermore, as secondary aim, we investigated how the implemented hygiene measures influenced the OSCE, including the interaction and communication between students and SPs.

### METHODS
### Study design

This study presents a cross-sectional study at the University of Tuebingen.

### The OSCE at the University of Tuebingen before the COVID-19 pandemic

The OSCE at the Medical School of Tuebingen takes place at the end of the third year of study. Nine clinical subjects participate in the OSCE: dermatology, general medicine, internal medicine, neurology, orthopaedics, paediatrics, psychosomatic medicine, radiology and surgery. The aim of the OSCE is the examination of students' clinical and communication skills. Clinical skills were represented

by performing a clinical examination or by interpreting a medical report. Communication skills were assessed by the way the students interacted with the patients, for example, by asking the patient for his or her name or introducing themselves using their name and function or by how they structured the communication with the patients. Typically, the students are examined at 16 examination stations representing basic tasks such as taking a patient's medical history, performing a clinical examination or interpreting medical findings (eg, an X-ray). The task at each station lasts 6 min with a 1 min break to change places. All interactive stations (medical history, physical examination) include SPs to provide a more realistic experience. Students are evaluated by discipline-specific examiners using an electronic standardised checklist consisting of 25 items on a tablet. Usually, about 180 medical students take part in the OSCE per semester.

### Implementation of the OSCE at the University of Tuebingen during the COVID-19 pandemic

During the pandemic, OSCEs took place in October 2020. The 7-day incidence rate at this time was 88 per 100 000 (https://www.tagblatt.de/Nachrichten/Live-Blog, 11 May 2021). Rapid testing had not yet been established as a means to test all participants for the coronavirus. Thus, the challenge was to adapt the OSCE to the university's hygiene measures including maintaining a social distance of at least 1.5 m during the OSCE and limiting the number of persons involved. As the number of participating students per run was limited to 16, the OSCE was extended to 3 days instead of 2 days, including six thematic blocks. The general structure of nine clinical subjects and 16 examination stations was retained. In addition, the examination time was kept to 6 min per station. However, the breaktime between stations was doubled to 2 min to accommodate performing hygienic procedures (eg, wiping all surfaces). All persons involved in the OSCE, including students, examiners, SPs and the coordination team, were required to wear face masks (minimum standard FFP1) for the total duration of the OSCE. As the OSCE always mirrors the clinical condition participants were not wearing disposable aprons. This also meant that several established stations had to be adapted because, as a result, several procedures were not possible (eg, examination of the larynx, facial nerve testing). For physical examinations, students were required to wear surgical gloves, dispose of them after each examination and put on a new pair for each new SP they met.

### Evaluation

The evaluation of the OSCE during the COVID-19 pandemic consisted of two parts: (1) a general evaluation (established standard) and (2) a COVID-19 pandemic-specific evaluation (resulting from changes due to COVID-19).

## General evaluation

The general evaluation included questions on the organisation of the OSCE as well as rating the preparation, content and difficulty of each examination station. The students were also asked if they felt confident in their ability to perform each examined task in a real-word situation. Finally, students were given the opportunity to comment on the OSCE in a free text section. All items were assessed by academic grades (1–6).

## The COVID-19 pandemic-specific evaluation

The specific evaluation in relation to COVID-19 referred to the effects of wearing face masks and was divided into the following topics: *interaction, non-verbal communication, face mask in general* and *face mask during OSCE*. An example of an item for *interaction* was 'The face mask influenced the interaction with the SP'. *Non-verbal communication* was assessed by the item 'They [students] made more use of the eyes in order to communicate with the SP'. An example of an item for *face mask in general* was represented by 'In general, I consider wearing a face mask as reasonable'. The topic *face mask during OSCE* was measured by items such as: 'Wearing a face mask affects taking the medical history/performing a physical examination'.

All items were rated on a 6-point Likert scale ranging from 1 ('completely') to 6 ('not at all').

## Procedure

The participating students first completed the OSCE and then completed the general and COVID-19-specific evaluations. In addition, SPs and examiners completed the general evaluation after the OSCE. Both groups rated the students' clinical skills, communication skills and empathy after completing each student's checklist in the 2 min break window.

## Patient and public involvement

Patients or the public were not involved in the design, or conduct, or reporting, or dissemination plans of our research.

## Data analysis

A sample size of 90 students was needed for analysis by using the standard statistical formula for sample size including study population=180, level of significance=0.95 and margin of error=0.5. The normal distribution of the data was confirmed by using the Kolmogorov-Smirnov test. Descriptive data, including mean values (M), standard deviations (SD), frequencies and percentages of relevant factors, were calculated. Any missing value was replaced with the mean value. The overall mean of missing values was estimated as 1.34%. Missing values were considered only if at least 80% of each item of the questionnaire had been completed. Using Little's missing completely at random test, it was confirmed that the data were missing randomly. The expectation-maximisation algorithm was used to input the missing data.[16] T-test for independent samples and analyses of variance were conducted to compare the results between the three groups including students, SPs and examiners. When data were not normally distributed, the Kruskal-Wallis test was performed to compare the results. Bonferroni correction was used to correct multiple testing.[17] The Statistical Package for the Social Sciences V.26.0 (IBM) was used for data analysis. The level of significance was set at $p < 0.05$.

## RESULTS

### Demographics

A total of 149 students completed the examination as well as the survey (RR=80.1%). Also participating in the survey were 32 SPs (RR=91.4%) and 59 examiners (RR=92.2%). All medical students were in their third year of study.

### General evaluation

Students evaluated the OSCE in regard to preparation at 2.37 (±0.52) and content and difficulty at 2.07 (±0.32). The confidence to perform the tasks in a real-world setting was rated 2.36 (±0.47). Both SPs and examiners rated the students' clinical skills as good (SPs: 1.83±0.79; examiners: 1.81±0.72, p>0.05). Similar results were found for students' communication skills (SPs: 1.72±0.77; examiners: 1.70±0.57, p>0.05). In regard to students' empathy with SPs, there was a significant effect (p<0.05) showing a higher rating by the examiners (1.51±0.55) when compared with the SPs (1.79±0.78).

### COVID-19-specific evaluation

#### Interaction and non-verbal communication

In regard to interaction and non-verbal communication, the results showed significant differences (p<0.001) for

**Table 1** Interaction and non-verbal communication

| Items* | Students | | SPs | | Examiners | | Statistics |
|---|---|---|---|---|---|---|---|
| | M | SD | M | SD | M | SD | P value |
| In general, interaction is affected by face masks. | 3.03 | 1.54 | 3.84 | 1.44 | 4.14 | 1.55 | <0.001 |
| Face masks have a negative impact on specific interactions with SPs. | 2.92 | 1.58 | 3.59 | 1.58 | 3.85 | 1.68 | <0.001 |
| Face masks lead to misunderstandings with SPs. | 3.85 | 1.59 | 5.43 | 1.04 | 4.82 | 1.38 | <0.001 |

*Items presented are from the students' questionnaire version; SPs/examiners had respective corresponding items (e.g. 'Wearing face masks interfered with the students conducting the anamnesis')
SP, standardised patient.

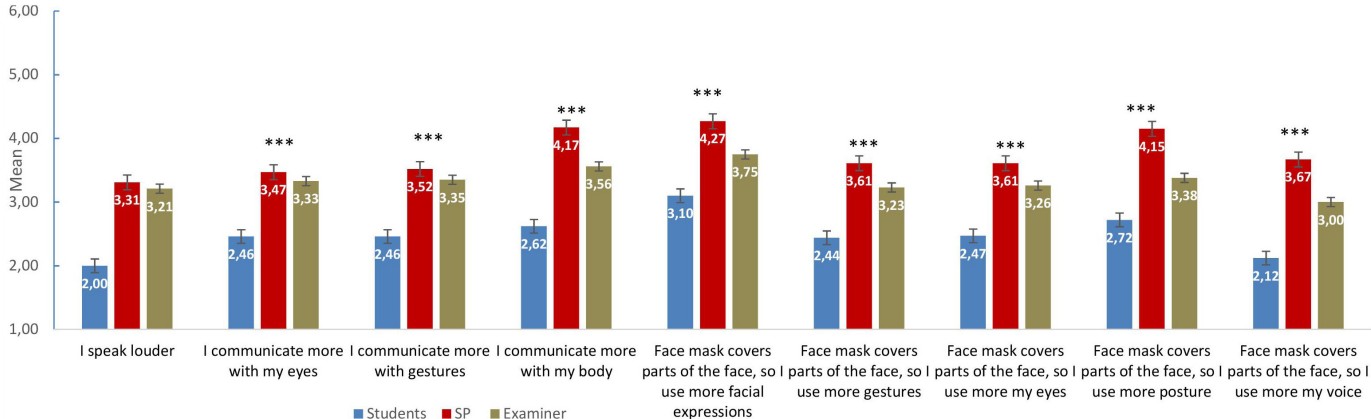

**Figure 1** Face mask and non-verbal communication in patient–physician communication. Items presented are from the students' questionnaire version; SPs/examiners had respective corresponding items (eg, 'the student spoke louder'). All items were rated on a 6-point Likert scale ranging from 1 ('completely') to 6 ('not at all'), ***p<0.001. SP, standardised patient.

the three groups, that is, students, SPs and examiners (see table 1 and figure 1). Students rated the single items of interaction and non-verbal communication with higher agreement than did the SPs and examiners. The students were significantly in greater agreement that wearing a face mask could lead to misunderstandings in comparison to the SPs and examiners, who both disagreed with the statement. Overall, compared with students and examiners, the SPs were in significant disagreement that face masks affected non-verbal communication by, for example, speaking more loudly or using more gestures.

### The use of face masks in general and in the OSCE

Students, SPs and examiners disagreed that wearing face masks had a negative impact on their well-being (range: 4.19–4.91). They all rated the face mask as useful in the OSCE (range: 1.52–1.79). However, students were more likely to agree that they would have preferred the option of participating in the OSCE the following semester without wearing a face mask (3.03±1.84). In regard to the content of the examination, SPs and examiners significantly disagreed that wearing a mask interfered with conducting an anamnesis. Furthermore, the SPs

significantly disagreed that wearing a face mask interfered with performing a physical examination during the OSCE. In addition, compared with SPs and examiners, students significantly disagreed that face masks affected communication during the OSCE. SPs and examiners significantly disagreed that face masks have negative impact on the students' grade. Please see table 2 and figure 2 for additional details.

### DISCUSSION

This study investigated the implementation of an OSCE during a pandemic. Regarding students' clinical and communication skills, SPs and examiners rated them as good, while the students themselves felt less confident about performing the required tasks in a real-world setting. When focusing on the changes required for appropriate hygienic measures, the SPs reported that face masks affected non-verbal communication, including speaking louder or using more eye contact, less in comparison to the examiners and students. Students, however, rated the face masks as having influenced the interaction and non-verbal communication with the SPs; therefore,

**Table 2** The use of face masks in general and in the OSCE

| Items* | Students | | SP | | Examiner | | Statistics |
|---|---|---|---|---|---|---|---|
| | M | SD | M | SD | M | SD | P value |
| Face masks, in general, are useful. | 1.58 | 1.08 | 2.67 | 2.13 | 1.92 | 1.34 | <0.001 |
| A face mask has a negative impact on my well-being. | 4.19 | 1.76 | 4.91 | 1.35 | 4.54 | 1.56 | >0.05 |
| Wearing a face mask causes physical symptoms (eg, headaches, getting less air, etc). | 4.68 | 1.63 | 5.47 | 0.95 | 4.79 | 1.68 | <0.05 |
| Wearing a face mask interfered with conducting the anamnesis. | 3.68 | 1.73 | 4.70 | 1.34 | 4.76 | 1.56 | <0.001 |
| Wearing a face mask interfered with performing a physical examination during the OSCE. | 3.81 | 1.65 | 5.33 | 1.04 | 4.44 | 1.80 | <0.001 |
| Face masks have a negative impact on the academic grade. | 4.03 | 1.64 | 4.93 | 1.86 | 5.00 | 1.58 | <0.001 |

*Items presented are from the students' questionnaire version; SPs/examiners had respective corresponding items (eg, 'Wearing face masks interfered with the students conducting the anamnesis').
OSCE, observed structured clinical examination; SP, standardised patient.

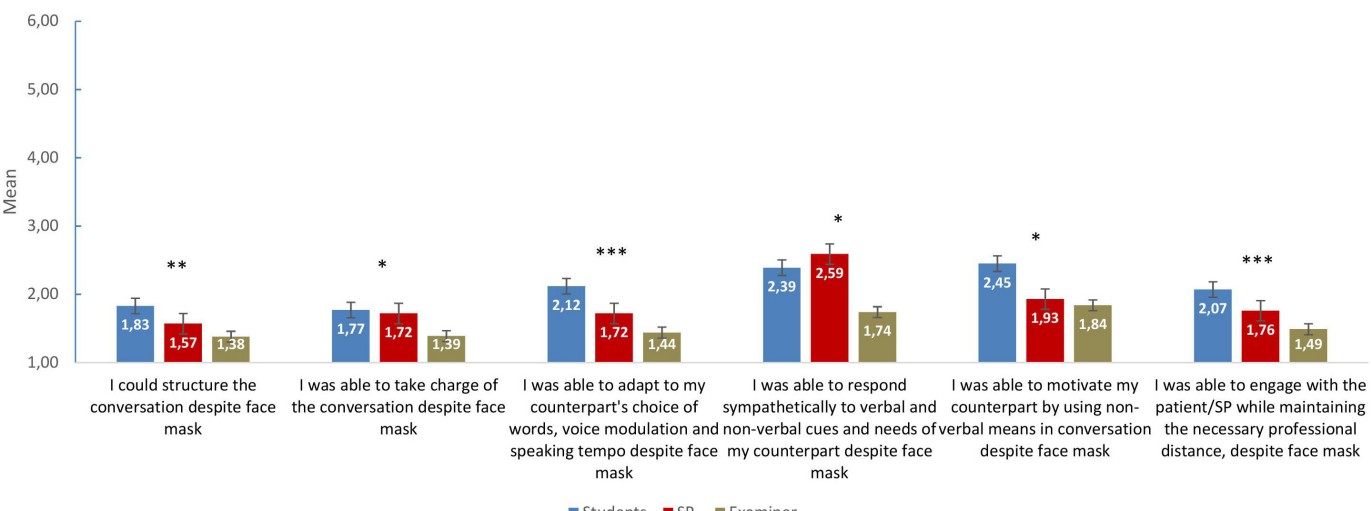

**Figure 2** Face mask and patient–physician communication in the OSCE. Items presented are from the students' questionnaire version; SPs/examiners had respective corresponding items (eg, 'the student could structure the conversation despite face mask'). All items were rated on a 6-point Likert scale ranging from 1 ('completely') to 6 ('not at all'). *P<0.05; **p>0.01; ***p<0.001. OSCE, observed structured clinical examination; SP, standardised patient.

they used more eye contact and spoke louder. In general, all participants found face masks to be a useful tool in the OSCE during the pandemic and reported that the masks did not interfere with performing the OSCE in general.

### Evaluation of an OSCE during a pandemic

When compared with previous results, that is, OSCE before COVID-19, the results indicated that students felt less prepared for this OSCE.[9] They also evaluated the content as more challenging and the difficulty as greater for the OSCE under the conditions required due to the pandemic.[9] Furthermore, they felt less confident about performing tasks such as taking a medical history or conducting a physical examination in a real-world setting, although the SPs and examiners rated their clinical and communication skills as good. The SPs and examiners' ratings of clinical and communication skills were similar to those of students' examination performances on previous OSCEs.[18 19] The reason may be that problematic circumstances during the semester (digital teaching, less emphasis on clinical skills) seem to have been counterbalanced by more lenient examiners who kept these difficulties in mind.

Furthermore, the OSCE presents a stressful examination situation in which many students feel distressed and anxious regardless of unusual circumstances such as those caused by the COVID-19 pandemic.[9 20 21] This year, the students experienced even greater distress during the OSCE since the academic courses to prepare them for it were transferred to online formats.[22] Thus, students might feel less confident about performing the tasks in a real-world situation.

### Face masks and interaction

In regard to interaction and non-verbal interaction, students were afraid that wearing face masks might affect their interaction with the SPs and that non-verbal communication might interfere with the SP during the tasks. Similar fears were also reported by psychiatrists who emphasised that the face masks contribute to a psychosocial barrier in the patient–physician interaction.[23] In addition, students agreed to speak louder when wearing a face mask. This phenomenon was confirmed by a previous study where a speech intelligibility assessment of protective face masks and air-purifying respirators was investigated.[24] However, examiners and SPs perceived the face masks as less disruptive than the students did. Examiners as clinicians may have become accustomed to them. Furthermore, SPs were specifically instructed in advance about how to communicate while wearing a face mask. Finally, all participants agreed that wearing face masks did not lead to any misunderstandings with the SPs, which was in contrast to previous studies that investigated face masks in relation to patient–physician communication.[25]

### Face masks in the OSCE

Wearing face masks was seen as useful in general, as well as in the OSCE, by students, SPs and examiners. This finding aligns with those of previous investigations of face masks.[25–27] Moreover, all participants disagreed that face masks might interfere with performing a physical examination or taking a medical history during the OSCE. While Campagne[28] emphasised that face masks contribute to distress in patient communication, the students in the present study reported that they did not affect the communication in the OSCE and that they were able to structure and take charge of the conversation despite face masks.[28] SPs and examiners agreed with the students. Medical students are well informed about COVID-19 and are aware of its consequences in academic and healthcare contexts.[22 29] Thus, the students might have felt confident about conducting a conversation while wearing a face mask in the OSCE. All participants also agreed that wearing face masks had no negative impact on the students' grades in the OSCE.

## Strengths and limitations

Based on the results, a strength of this study includes the finding that an in-person OSCE can be effectively performed even during a severe pandemic if it is done in a thoughtful and structured format, keeping both the educational objectives as well as proper infection control practices in mind. Moreover, best to our knowledge, the study is the first investigation of the influence of face masks regarding communication and interaction in an OSCE. Furthermore, the perspective of all people involved in the OSCE was assessed.

A limitation of this study is the higher level of stress in medical students during the OSCE as it is an examination and stress has complex interactions with students' performance and the way how they judge situations.[30] Additionally, the study was conducted at only one medical faculty; thus, the results cannot be generalised. Furthermore, the cross-sectional study design only allows to capture the status of the current situation and a longitudinal study design is needed to determine possible causal effects. Despite its limitations, we believe that this study contributes to the knowledge about how to conduct effective OSCEs and possibly other student assessments in unusual circumstances such as the COVID-19 pandemic, as well as the challenges that must be considered.

## CONCLUSION

Wearing face masks did not interfere with student performance regarding clinical and communication skills in the OSCE. Students felt less prepared for the OSCE as 'refresher' courses were missing before the OSCE took place, and academic courses were transferred to online formats where relevant teaching elements, for example, of physical examination, could be lost.[22] Again, these results showed that students were aware of the circumstances and consequences of COVID-19. In conclusion, an in-person OSCE, even in the midst of a global pandemic, is feasible and acceptable to both students and faculty. Furthermore, students are willing to fulfil the hygienic measurements like wearing a face mask. However, the clinical educators are prompted to create and implement 'refresher' courses in order to deliver good preparation for students and to compensate the missing face-to-face courses before the OSCE. Future research might also focus on the students' grades and performances in the OSCE during the pandemic, including a comparison with prepandemic OSCE results before the restrictions and changes required to prevent the spread of COVID-19.[9]

**Acknowledgements** We thank Lea Herschbach for her support as study assistant.

**Contributors** AHW and TL were responsible for the design and conduct of the study, as well as acquisition, analysis and interpretation of data. AHW and TL drafted the first version of the manuscript. AHW also took the role of the guarantor for the study. RSE and ABN were involved in data analyses and interpretation and revised the manuscript critically. SZ made substantial contributions to the study design and revised the manuscript critically. All authors approved the final version of the manuscript and agreed to be accountable for all aspects of the work.

**Funding** We acknowledge the support with financing publication fees by 'Deutsche Forschungsgemeinschaft' and Open Access Publishing Fund of the University of Tuebingen.

**Competing interests** None declared.

**Patient and public involvement** Patients and/or the public were not involved in the design, or conduct, or reporting, or dissemination plans of this research.

**Patient consent for publication** Obtained.

**Ethics approval** The study received ethics approval from the Ethics Committee of Tuebingen Medical Faculty (reference number: 314/2020B02). Participation was voluntary, and students were not reimbursed or rewarded in any way for participating. All participants provided their written informed consent, and all of their responses and data were kept anonymous.

**Provenance and peer review** Not commissioned; externally peer reviewed.

**Data availability statement** Data are available upon reasonable request.

**ORCID iDs**
Teresa Loda http://orcid.org/0000-0003-1450-1757
Rebecca Sarah Erschens http://orcid.org/0000-0002-4433-9378
Anne Herrmann-Werner http://orcid.org/0000-0003-2413-7047

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
