## [Reviewer comments · BMJ Open]

ARTICLE DETAILS

TITLE (PROVISIONAL)	An original article: Perspectives, benefits and challenges of a live OSCE during the COVID-19 Pandemic in a cross-sectional study
AUTHORS	Loda, Teresa; Erschens, Rebecca; Nevins, Andrew B; Zipfel, Stephan; Herrmann-Werner, Anne

VERSION 1 – REVIEW

REVIEWER	Haque, Mainul National Defense University of Malaysia, Faculty of Medicine and Defense Health, Pharmacology
REVIEW RETURNED	15-Jan-2022

GENERAL COMMENTS	Good work. Interesting paper. Methods Section It is difficult to understand. How do calculate 149 sample size? What was the sampling method? What was the study period and Are? Please add the full address of the Ethical Approval Committee with date of Reference number. "A limitation of this study is the higher level of stress in medical students during the OSCE as it is an examination. " do it is a limitation? Similarly "This study contributes to the knowledge about how to conduct effective OSCEs and possibly other student assessments in unusual circumstances such as the COVID-19 pandemic, as well as the challenges that must be considered ": Please add the limitation of the cross-sectional study. Please add what new findings this research brings in or add to medical education in the conclusion section. Please add your recommendation containing future implications of this research
--

REVIEWER	Patel, Nimesh Queen Mary University of London, Centre for Medical Education
REVIEW RETURNED	24-Jan-2022

GENERAL COMMENTS	The study seeks to investigate the impact of infection control
--

	measures on practical skills and face masks on candidate-patient interactions. Cohort size = 149 in year 3. Abstract: When was the OSCE conducted? second sentence of "Primary and Secondary outcome measures" is unclear - the general evaluation of "structure and content" of what? what are the primary measures and what are the secondary measures? Likert scale definition needs to be provided to understand results in abstract - 1 = xxx and 6 = xxx. How has the conclusion of "students felt less prepared than under non-pandemic circumstances" been inferred. Does your medical school have OSCEs in the earlier years so has this been their third OSCE, second OSCE or first?? Introduction: References 12 and 13 more related to dental OSCEs and would be better to include references relating to medical OSCEs. Aim: different to that described in abstract. "authentic context" - do you mean validity? Implementation: did participants at the OSCE have to wear disposable aprons? The rationale for replacing missing data points with the mean value should be provided and the number of instances this happened. Page 8, Line 32: "All items were rated..." should be moved to the end of "General Evaluation" section. full stop missing - page 7, line 9. Evaluation of an OSCE during pandemic: second to last sentence has a reference missing. Face masks and interaction: If SPs were instructed how to communicate with a face mask, were students instructed too? The aims and objectives of the study have not been clearly articulated and need to be revised for more consistency throughout. The primary and secondary outcome measures need greater clarity - it is clear there are both primary and secondary outcome measures but it requires reading between the lines.
--	---

VERSION 1 – AUTHOR RESPONSE

Reviewer: 1

Dr. Mainul Haque, National Defense University of Malaysia, Faculty of Medicine and Defense Health

Comments to the Author:

Dear Author

Good work.

Interesting paper.

Thank you for your appreciation.

Methods Section

It is difficult to understand.
How do calculate 149 sample size?
What was the sampling method?
What was the study period and Are?

Thank you for your questions regarding sampling and study period. We have added respective comments in the methods section. Unfortunately, we do not understand the final part of the sentence "... and Are?". If it was an important point, please do not hesitate to come back to us.

Please add the full address of the Ethical Approval Committee with date of Reference number.

It seems a bit unusual to add the complete address within the manuscript. Thus, we have added the full address in a commentary and the editor can decide if it should be included in the actual manuscript or not. We hope this is an acceptable way for you.

"A limitation of this study is the higher level of stress in medical students during the OSCE as it is an examination." do it is a limitation?

Similarly "This study contributes to the knowledge about how to conduct effective OSCEs and possibly other student assessments in unusual circumstances such as the COVID-19 pandemic, as well as the challenges that must be considered ":

Please add the limitation of the cross-sectional study.

Please add what new findings this research brings in or add to medical education in the conclusion section.

Please add your recommendation containing future implications of this research

Thank you for your comments. Stress has complex interactions with performance and the way how people judge situations. Thus, we do consider it a limitation. We have added some more explanation to make this point clearer. We have also added the limitation of cross-sectional studies. In the conclusions section, we have added the additional value of our research as well as some more further implications.

Reviewer: 2

Dr. Nimesh Patel, Queen Mary University of London Comments to the Author:

The study seeks to investigate the impact of infection control measures on practical skills and face masks on candidate-patient interactions. Cohort size = 149 in year 3.

Abstract: When was the OSCE conducted? second sentence of "Primary and Secondary outcome measures" is unclear - the general evaluation of "structure and content" of what? what are the primary measures and what are the secondary measures? Likert scale definition needs to be provided to understand results in abstract - 1 = xxx and 6 = xxx. How has the conclusion of "students felt less prepared than under non-pandemic circumstances" been inferred. Does your medical school have OSCEs in the earlier years so has this been their third OSCE, second OSCE or first??

Thank you for your remarks on the abstract. We absolutely agree with your comments – partly the potential for misunderstanding may have arisen due to word limitations. Thus, we tried to accommodate all points mentioned by

- Adding the date as well as the number of the OSCE examined
- Rephrasing the sentence about primary and secondary measures
- Defining the values of our Likert scale
- Explaining the conclusion about students' feeling of preparedness

We hope we have satisfyingly addressed all concerns raised regarding the abstract.

Introduction: References 12 and 13 more related to dental OSCEs and would be better to include references relating to medical OSCEs.

Thank you for this suggestion. Although we do completely agree that human and dental medicine are different stories, we still believe that the references used have their place in our introduction as they show first attempts how to implement an OSCE in the pandemic when searching for relevant literature

in 2020. However, we searched again for literature and inserted new references regarding human medicine.

Aim: different to that described in abstract. "authentic context" - do you mean validity?

Thank you for your comment. No, we meant how well the students could manage the OCES's tasks in real life. We changed the aim to make it clearer.

Implementation: did participants at the OSCE have to wear disposable aprons?

Thank you for your question. We are aware of the fact that PPE includes disposable aprons, too. However, in Germany, aprons only had to be worn when doing a swab or in direct infectious contact. As our OSCE always mirrors our clinical conditions, students were not wearing aprons. We have included an explanatory sentence when talking about masks and gloves.

The rationale for replacing missing data points with the mean value should be provided and the number of instances this happened.

Thank you for your comment. We included an explanation on the rationale as well as the number of instances as required in the statistical analysis.

Page 8, Line 32: "All items were rated..." should be moved to the end of "General Evaluation" section.

Thank you for your suggestion – we have moved the sentence accordingly.

full stop missing - page 7, line 9.

Please excuse the omission – we have added the full stop.

Evaluation of an OSCE during pandemic: second to last sentence has a reference missing.

Thank you for your comment. We are sorry for this mistake and inserted the reference missing.

Face masks and interaction: If SPs were instructed how to communicate with a face mask, were students instructed too?

Thank you for your comment. The students were not instructed how to communicate with a face mask. They only had a rehearsal one week before the OSCE where they could practice how to communicate with face mask or how to do the physical examination with surgical gloves. We inserted a sentence into the methods to make it clearer.

The aims and objectives of the study have not been clearly articulated and need to be revised for more consistency throughout. The primary and secondary outcome measures need greater clarity - it is clear there are both primary and secondary outcome measures but it requires reading between the lines.

Thank you for your valuable comment and highlighting general issues you experienced with the paper. We have tried to sharpen the issues around aims and objectives as well as outcome measures and hope that it is now clearer.

VERSION 2 – REVIEW

REVIEWER	Patel, Nimesh Queen Mary University of London, Centre for Medical Education
REVIEW RETURNED	21-Mar-2022
GENERAL COMMENTS	The manuscript is now better presented than the original version and clearly articulates the aims and objectives consistently throughout. A very interesting piece of work.